# Bayesian Framework for Gradient Leakage

**Mislav Balunović, Dimitar I. Dimitrov, Robin Staab, Martin Vechev**
Department of Computer Science
ETH Zurich
`{mislav.balunovic,dimitar.dimitrov,`
` robin.staab,martin.vechev}@inf.ethz.ch`

## Abstract

Federated learning is an established method for training machine learning models without sharing training data. However, recent work has shown that it cannot guarantee data privacy as shared gradients can still leak sensitive information. To formalize the problem of *gradient leakage*, we propose a theoretical framework that enables, for the first time, analysis of the Bayes optimal adversary phrased as an optimization problem. We demonstrate that existing leakage attacks can be seen as approximations of this optimal adversary with different assumptions on the probability distributions of the input data and gradients. Our experiments confirm the effectiveness of the Bayes optimal adversary when it has knowledge of the underlying distribution. Further, our experimental evaluation shows that several existing heuristic defenses are not effective against stronger attacks, especially early in the training process. Thus, our findings indicate that the construction of more effective defenses and their evaluation remains an open problem.

## 1 Introduction

Federated learning (McMahan et al., 2017) has become a standard paradigm for enabling users to collaboratively train machine learning models. In this setting, clients compute updates on their own devices, send the updates to a central server which aggregates them and updates the global model. Because user data is not shared with the server or other users, this framework should, in principle, offer more privacy than simply uploading the data to a server. However, this privacy benefit has been increasingly questioned by recent works (Melis et al., 2019; Zhu et al., 2019; Geiping et al., 2020; Yin et al., 2021) which demonstrate the possibility of reconstructing the original input from shared gradient updates. The reconstruction works by optimizing a candidate image with respect to a loss function that measures the distance between the shared and candidate gradients. The attacks typically differ in their loss function, their regularization, and how they solve the optimization problem. Importantly, the success of these attacks raises the following key questions: (i) *what is the theoretically worst-case attack?*, and (ii) *how do we evaluate defenses against gradient leakage?*

**This work** In this work, we study and address these two questions from both theoretical and practical perspective. Specifically, we first introduce a theoretical framework which allows us to measure the expected risk an adversary has in reconstructing an input, given the joint probability distribution of inputs and their gradients. We then analyze the Bayes optimal adversary, which minimizes this risk and show that it solves a specific optimization problem involving the joint distribution. Further, we phrase existing attacks (Zhu et al., 2019; Geiping et al., 2020; Yin et al., 2021) as approximations of this optimal adversary, where each attack can be interpreted as implicitly making different assumptions on the distribution of gradients and inputs, in turn yielding different loss functions for the optimization. In our experimental evaluation, we compare the Bayes optimal adversary with other attacks, those which do not leverage the probability distribution of gradients, and we find that the Bayes optimal adversary performs better, as explained by the theory. We then turn to practical evaluation and experiment with several recently proposed defenses (Sun et al., 2021; Gao et al., 2021; Scheliga et al., 2021) based on different heuristics and demonstrate that they do not protect from gradient leakage against stronger attacks that we design specifically for each defense. Interestingly, we find that models are especially vulnerable to attacks early in training, and thus we advocate that defense evaluation should take place during and not only at the end of training. Overall, our

findings suggest that creation of effective defenses and their evaluation is a challenging problem, and that our insights and contributions can substantially advance future research in the area.

**Main contributions**   Our main contributions are:

- Formulation of the gradient leakage problem in a Bayesian framework which enables phrasing Bayes optimal adversary as an optimization problem.

- Interpretation of several prior attacks as approximations of the Bayes optimal adversary, each using different assumptions for the distributions of inputs and their gradients.

- Practical implementation of the Bayes optimal adversary for several defenses, showing higher reconstruction success than previous attacks, confirming our theoretical results. We make our code publicly available at `https://github.com/eth-sri/bayes-framework-leakage`.

- Evaluation of several existing heuristic defenses demonstrating that they do not effectively protect from strong attacks, especially early in training, thus suggesting better ways for evaluating defenses in this space.

## 2   RELATED WORK

We now briefly survey some of the work most related to ours.

**Federated Learning**   Federated learning (McMahan et al., 2017) was introduced as a way to train machine learning models in decentralized settings with data coming from different user devices. This new form of learning has caused much interest in its theoretical properties (Konečný et al., 2016a) and ways to improve training efficiency (Konečný et al., 2016b). More specifically, besides decentralizing the computation on many devices, the fundamental promise of this approach is privacy, as the user data never leaves their devices.

**Gradient Leakage Attacks**   Recent work (Zhu et al., 2019; Geiping et al., 2020) has shown that such privacy assumptions, in fact, do not hold in practice, as an adversarial server can reliably recover an input image from gradient updates. Given the gradient produced from an image $x$ and a corresponding label $y$, they phrase the attack as a minimization problem over the $\ell_2$-distance between the original gradient and the gradient of a randomly initialized input image $x'$ and label $y'$:

$$(x^*, y^*) = \underset{(x', y')}{\arg\min} ||\nabla l(h_\theta(x), y) - \nabla l(h_\theta(x'), y')||_2 \tag{1}$$

Here $h_\theta$ is a neural network and $l$ denotes a loss used to train the network, usually cross-entropy. Follow-up works improve on these results by using different distance metrics such as cosine similarity and input-regularization (Geiping et al., 2020), smarter initialization (Wei et al., 2020), normalization (Yin et al., 2021), and others (Mo et al., 2021; Jeon et al., 2021). A significant improvement proposed by Zhao et al. (2020) showed how to recover the target label $y$ from the gradients alone, reducing Eq. (1) to an optimization over $x'$ only. In Section 4 we show how these existing attacks can be interpreted as different approximations of the Bayes optimal adversary.

**Defenses**   In response to the rise of privacy-violating attacks on federated learning, many defenses have been proposed (Abadi et al., 2016; Sun et al., 2021; Gao et al., 2021). Except for DP-SGD (Abadi et al., 2016), a version of SGD with clipping and adding Gaussian noise, which is differentially private, they all provide none or little theoretical privacy guarantees. This is partly due to the fact that no mathematically rigorous attacker model exists, and defenses are empirically evaluated against known attacks. This also leads to a wide variety of proposed defenses: Soteria (Sun et al., 2021) prunes the gradient for a single layer, ATS (Gao et al., 2021) generates highly augmented input images that train the network to produce non-invertible gradients, and PRECODE (Scheliga et al., 2021) uses a VAE to hide the original input. Here we do not consider defenses that change the communication and training protocol (Lee et al., 2021; Wei et al., 2021).

## 3 BACKGROUND

Let $\mathcal{X} \subseteq \mathbb{R}^d$ be an input space, and let $h_\theta : \mathcal{X} \to \mathcal{Y}$ be a neural network with parameters $\theta$ classifying an input $x$ to a label $y$ in the label space $\mathcal{Y}$. We assume that inputs $(x, y)$ are coming from a distribution $\mathcal{D}$ with a marginal distribution $p(x)$. In standard federated learning, there are $n$ clients with loss functions $l_1, ..., l_n$, who are trying to jointly solve the optimization problem:

$$\min_\theta \frac{1}{n} \sum_{i=1}^n \mathbb{E}_{(x,y)\sim\mathcal{D}} \left[ l_i(h_\theta(x), y) \right].$$

To ease notation, we will assume a single client throughout the paper, but the same reasoning can be applied to the general $n$-client case. Additionally, each client could have a different distribution $\mathcal{D}$, but the approach is again easy to generalize to this case. In a single training step, each client $i$ first computes $\nabla_\theta l_i(h_\theta(x_i), y_i)$ on a batch of data $(x_i, y_i)$, then sends these to the central server that performs a gradient descent step to obtain the new parameters $\theta' = \theta - \frac{\alpha}{n} \sum_{i=1}^n \nabla_\theta l_i(h_\theta(x_i), y_i)$, where $\alpha$ is a learning rate. We will consider a scenario where each client reports, instead of the true gradient $\nabla_\theta l_i(h_\theta(x_i), y_i)$, a noisy gradient $g$ sampled from a distribution $p(g|x)$, which we call a defense mechanism. The purpose of the defense mechanism is to add enough noise to hide the sensitive user information from the gradients while retaining high enough informativeness so that it can be used for training. Thus, given some noisy gradient $g$, the central server would update the parameters as $\theta' = \theta - \alpha g$ (assuming $n = 1$ as mentioned above). Typical examples of defenses used in our experiments in Section 6, each inducing $p(g|x)$, include adding Gaussian or Laplacian noise to the original gradients, as well as randomly masking some components of the gradient. This setup also captures the common DP-SGD defense (Abadi et al., 2016) where $p(g|x)$ is a Gaussian centered around the clipped true gradient. Naturally, $p(x)$ and $p(g|x)$ together induce a joint distribution $p(x, g)$. Note that a network that has no defense corresponds to the distribution $p(g|x)$, which is concentrated only on the true gradient at $x$.

## 4 BAYESIAN ADVERSARIAL FRAMEWORK

Here we describe our theoretical framework for gradient leakage in federated learning. As we show, our problem setting captures many commonly used defenses such as DP-SGD (Abadi et al., 2016) and many attacks from prior work (Zhu et al., 2019; Geiping et al., 2020).

**Adversarial risk** We first define the adversarial risk for gradient leakage and then derive the Bayes optimal adversary that minimizes this risk. The adversary can only observe the gradient $g$ and tries to reconstruct the input $x$ that produced $g$. Formally, the adversary is a function $f : \mathbb{R}^k \to \mathcal{X}$ mapping gradients to inputs. Given some $(x, g)$ sampled from the joint distribution $p(x, g)$, the adversary outputs the reconstruction $f(g)$ and incurs loss $\mathcal{L}(x, f(g))$, which is a function $\mathcal{L} : \mathcal{X} \times \mathcal{X} \to \mathbb{R}$. Typically, we will consider a binary loss that evaluates to 0 if the adversary's output is close to the original input, and 1 otherwise. If the adversary wants to reconstruct the *exact* input $x$, we can define the loss to be $\mathcal{L}(x, x') := 1_{x \neq x'}$, denoting with 1 the indicator function. If the adversary only wants to get to some $\delta$-neighbourhood of the input $x$ in the input space, a more appropriate definition of the loss is $\mathcal{L}(x, x') := 1_{d(x,x')>\delta}$. In this section, we will assume that the distance $d$ is the $\ell_2$-distance, but the approach can be generalized to other notions as well. This definition is well suited for image data, where $\ell_2$ distance captures our perception of visual closeness, and for which the adversary can often obtain a reconstruction that is very close to the original image, both visually and in the $\ell_2$ space. Note that if we let $\delta$ approach 0 in our second definition of the loss, we essentially recover the first loss. We can now define the risk $R(f)$ of the adversary $f$ as

$$R(f) := \mathbb{E}_{x,g} \left[ \mathcal{L}(x, f(g)) \right] = \mathbb{E}_{x\sim p(x)} \mathbb{E}_{g\sim p(g|x)} \left[ \mathcal{L}(x, f(g)) \right].$$

**Bayes optimal adversary** We consider white-box adversary who knows the joint distribution $p(x, g)$, as opposed to the weaker alternative based on security through obscurity, where adversary does not know what defense is used, and therefore does not have a good estimate of $p(x, g)$. We also do not consider adversaries that can exploit other vulnerabilities in the system to obtain extra information. Let us consider the second definition of the loss for which $\mathcal{L}(x, f(g)) := 1_{||x - f(g)||_2 > \delta}$.

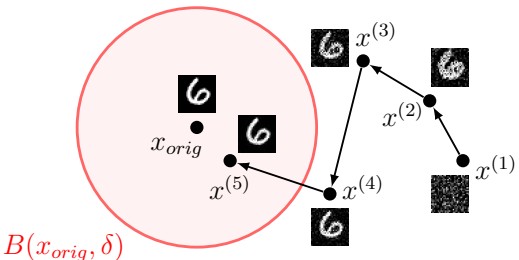

Figure 1: Example of a gradient leakage attack. Bayes optimal adversary randomly initializes image $x^{(1)}$ and then optimizes for the image with the highest $\log p(g|x) + \log p(x)$ in its $\delta$-neighborhood. The adversary has loss 1 if the final reconstruction is outside the ball $B(x_{orig}, \delta)$, and 0 otherwise.

We can then rewrite the definition of the risk as follows:

$$
\begin{aligned}
R(f) &= \mathbb{E}_{x,g}\left[\mathcal{L}(x, f(g))\right] \\
&= \mathbb{E}_g \mathbb{E}_{x|g}\left[1_{||x-f(g)||_2 > \delta}\right] \\
&= \mathbb{E}_g \int_{\mathcal{X}} p(x|g) \cdot 1_{||x-f(g)||_2 > \delta} \, dx \\
&= \mathbb{E}_g \int_{\mathcal{X}\backslash B(f(g),\delta)} p(x|g) \, dx \\
&= 1 - \mathbb{E}_g \int_{B(f(g),\delta)} p(x|g) \, dx.
\end{aligned}
$$

Here $B(f(g), \delta)$ denotes the $\ell_2$-ball of radius $\delta$ around $f(g)$. Thus, an adversary which wants to minimize their risk has to maximize $\mathbb{E}_g \int_{B(f(g),\delta)} p(x|g) \, dx$, meaning that the adversarial function $f$ can be defined as $f(g) := \arg\max_{x_0} \int_{B(x_0,\delta)} p(x|g) \, dx$. Intuitively, the adversary predicts $x_0$ which has the highest likelihood that one of the inputs in its $\delta$-neighborhood was the original input that produced gradient $g$. Note that if we would let $\delta \to 0$, then $f(g) \to \arg\max_x p(x|g)$, which would be the solution for the loss that requires the recovered input to exactly match the original input. As we do not have the closed form for $p(x|g)$, it can be rewritten using Bayes' rule:

$$
\begin{aligned}
f(g) &= \arg\max_{x_0 \in \mathcal{X}} \int_{B(x_0,\delta)} p(x|g) \, dx \\
&= \arg\max_{x_0 \in \mathcal{X}} \int_{B(x_0,\delta)} \frac{p(g|x)p(x)}{p(g)} \, dx \\
&= \arg\max_{x_0 \in \mathcal{X}} \int_{B(x_0,\delta)} p(g|x)p(x) \, dx \\
&= \arg\max_{x_0 \in \mathcal{X}} \left[\log \int_{B(x_0,\delta)} p(g|x)p(x) \, dx\right]
\end{aligned} \tag{2}
$$

Computing the optimal reconstruction now requires evaluating both the input prior $p(x)$ and the conditional probability $p(g|x)$, which is determined by the used defense mechanism. Given these two ingredients, Eq. (2) then provides us with a way to compute the output of the Bayes optimal adversary by solving an optimization problem involving distributions $p(x)$ and $p(g|x)$.

**Approximate Bayes optimal adversary**   While Eq. (2) provides a formula for the optimal adversary in the form of an optimization problem, using this adversary for practical reconstruction is difficult due to three main challenges: (i) we need to know the exact prior distribution $p(x)$, (ii) in general computing the integral over the $\delta$-ball around $x_0$ is intractable, and (iii) we need to solve the optimization problem over $\mathcal{X}$. However, we can address each of these challenges by introducing

| Attack | Prior $p(x)$ | Conditional $p(g|x)$ |
|---|---|---|
| DLG (Zhu et al., 2019) | Uniform | Gaussian |
| Inverting Gradients (Geiping et al., 2020) | TV | Cosine |
| GradInversion (Yin et al., 2021) | TV + Gaussian + DeepInv | Gaussian |

Table 1: Several existing attacks can be interpreted as instances of our Bayesian framework. We show prior and conditional distribution for corresponding losses that each attack uses.

appropriate approximations. We first apply Jensen's inequality to the logarithm function:

$$\max_{x_0 \in \mathcal{X}} \left[ \log C \int_{B(x_0,\delta)} p(g|x)p(x)\, dx \right] \geq \max_{x_0 \in \mathcal{X}} \left[ C \int_{B(x_0,\delta)} (\log p(g|x) + \log p(x))\, dx \right].$$

Here $C = 1/\delta^d$ is a normalization constant. For image data, we approximate $\log p(x)$ using the total variation image prior, which has already worked well for Geiping et al. (2020). Alternatively, we could estimate it from data using density estimation models such as PixelCNN (van den Oord et al., 2016) or Glow (Kingma & Dhariwal, 2018). We then approximate the integral over the $\delta$-ball via Monte Carlo integration by sampling $k$ points $x_1, ..., x_k$ uniformly in the ball to obtain the objective:

$$\frac{1}{k} \sum_{i=1}^{k} \log p(g|x_i) + \log p(x_i). \tag{3}$$

Finally, as the objective is differentiable, we can use gradient-based optimizer such as Adam (Kingma & Ba, 2015), and obtain the attack in Algorithm 1. Fig. 1 shows a single run of this adversary which is initialized randomly, and gets closer to the original image at every step.

**Existing attacks as approximations of the Bayes optimal adversary** We now describe how existing attacks can be viewed as different approximations of the Bayes optimal adversary. Recall that the optimal adversary $f$ searches for the $\delta$-ball with the maximum value for the integral of $\log p(g|x) + \log p(x)$. Next we show previously proposed attacks can in fact be recovered by plugging in different approximations for $\log p(g|x)$

---

**Algorithm 1** Approximate Bayes optimal adversary

$x^{(1)} \leftarrow$ attack_init()
**for** $i = 1$ **to** $m - 1$ **do**
  Sample $x_1, ..., x_k$ uniformly from $B(x^{(i)}, \delta)$
  $x^{(i+1)} \leftarrow x^{(i)} + \alpha \nabla_x \frac{1}{k} \sum_{i=1}^{k} \log p(g|x_i) + \log p(x_i)$
**end for**
**return** $x^{(m)}$

---

and $\log p(x)$ into Algorithm 1, estimating the integral using $k = 1$ samples located at the center of the ball. For example, suppose that the defense mechanism adds Gaussian noise to the gradient, which corresponds to the conditional probability $p(g|x) = \mathcal{N}(\nabla_\theta l(h_\theta(x), y), \sigma^2 I)$. This implies $\log p(g|x) = C - \frac{1}{2\sigma^2}||g - \nabla_\theta l(h_\theta(x), y)||_2^2$, where $C$ is a constant. Assuming a uniform prior (where $\log p(x)$ is constant), the problem in Eq. (2) simplifies to minimizing the $\ell_2$ distance between $g$ and $\nabla_\theta l(h_\theta(x), y)$, recovering exactly the optimization problem solved by Deep Leakage from Gradients (DLG) (Zhu et al., 2019). Inverting Gradients (Geiping et al., 2020) uses a total variation (TV) image prior for $\log p(x)$ and cosine similarity instead of $\ell_2$ to measure similarity between gradients. Cosine similarity corresponds to the distribution $\log p(g|x) = C - \frac{g^T \nabla_\theta l(h_\theta(x), y)}{||g||_2 ||\nabla_\theta l(h_\theta(x), y)||_2}$ which also requires the support of the distribution to be bounded. This happens, e.g., when gradients are clipped. GradInversion (Yin et al., 2021) introduces a more complex prior based on a combination of the total variation, $\ell_2$ norm of the image, and a DeepInversion prior while using $l_2$ norm to measure distance between gradients, which corresponds to Gaussian noise, as observed before. Note that we can work with unnormalized densities, as multiplying by the normalization constant does not change the argmax in Eq. (2). We summarize these observations in Table 1, showing the respective conditional and prior probabilities for each of the discussed attacks.

## 5 BREAKING EXISTING DEFENSES

In this section we provide practical attacks against three recent defenses, showing they are not able to withstand stronger adversaries early in training. While in Section 4 we have shown that Bayes optimal adversary is the optimal attack for any defense (each with different $p(g|x)$), we now show how practical approximations of this attack can be used to attack existing defenses (more details in Appendix A.5). In Section 6 we show experimental results obtained using our attacks, demonstrating the need to evaluate defenses against closest possible approximation to the optimal attack.

**Soteria** The Soteria defense (Sun et al., 2021) perturbs the intermediate representation of the input at a chosen defended layer $l$ of the attacked neural network $H : \mathbb{R}^{n_0} \to \mathbb{R}^{n_L}$ with $L$ layers of size $n_1, \ldots, n_L$ and input size $n_0$. Let $X$ and $X' \in \mathbb{R}^{n_0}$ denote the original and reconstructed images on $H$ and $h_{i,j} : \mathbb{R}^{n_i} \to \mathbb{R}^{n_j}$ denote the function between the input of the $i^{\text{th}}$ layer of $H$ and the output of the $j^{\text{th}}$. For the chosen layer $l$, Sun et al. (2021) denotes the inputs to that layer for the images $X$ and $X'$ with $r = h_{0,l-1}(X)$ and $r' = h_{0,l-1}(X')$, respectively, and aims to solve

$$\max_{r'} ||X - X'||_2 \text{ s.t. } ||r - r'||_0 \leq \epsilon. \tag{4}$$

Intuitively, Eq. (4) is searching for a minimal perturbation of the input to layer $l$ that results in maximal perturbation of the respective reconstructed input $X'$. Despite the optimization being over the intermediate representation $r'$, for an attacker who observes neither $r$ nor $r'$ the defense amounts to a perturbed gradient at layer $l$. In particular let $\nabla W = \{\nabla W_1, \nabla W_2, \ldots \nabla W_L\}$ be the set of gradients for the variables in the different layers of $H$. Sun et al. (2021) first solves Eq. (4) to obtain $r'$. It afterwards uses this $r'$ to generate a perturbed gradient at layer $l$ denoted with $\nabla W'_l$. Notably $r'$ is not propagated further through the network and hence the gradient perturbation stays local to the defended layer $l$. Hence, to defend the data $X$, Soteria's clients send the perturbed set of gradients $\nabla W' = \{\nabla W_1, \ldots, \nabla W_{l-1}, \nabla W'_l, \nabla W_{l+1}, \ldots \nabla W_L\}$ in place of $\nabla W$. Sun et al. (2021) show that Soteria is safe against several attacks from prior work (Zhu et al., 2019; Geiping et al., 2020).

In this work, we propose to circumvent this limitation by dropping the perturbed gradients $\nabla W'_l$ from $\nabla W'$ to obtain $\nabla W^* = \{\nabla W_1, \ldots, \nabla W_{l-1}, \nabla W_{l+1}, \ldots \nabla W_L\}$. As long as $\nabla W^*$ contains enough gradients, this allows an attacker to compute an almost perfect reconstruction of $X$. Note that the attacker does not know which layer is defended, but can simply run the attack for all.

**Automated Transformation Search** The Automatic Transformation Search (ATS) (Gao et al., 2021) attempts to hide sensitive information from input images by augmenting the images during training. The key idea is to score sequences of roughly 3 to 6 augmentations from AutoAugment library (Cubuk et al., 2019) based on the ability of the trained network to withstand gradient-based reconstruction attacks and the overall network accuracy. Similarly to Soteria, Gao et al. (2021) also demonstrate that ATS is safe against attacks proposed by Zhu et al. (2019) and Geiping et al. (2020).

In this work we show that, even though the ATS defense works well in later stages of training, in initial communication rounds we can easily reconstruct the input images using Geiping et al. (2020)'s attack. Our experiments in Section 6 on the network architecture and augmentations introduced in Gao et al. (2021) indicate that an attacker can successfully extract large parts of the input despite heavy image augmentation. The main observation is that the gradients of a randomly initialized network allow easier reconstruction independent of the input. Only after training, which changes gradient distribution, can defenses such as ATS empirically defend against gradient leakage.

**PRECODE** PRECODE (Scheliga et al., 2021) is a proposed defense which inserts a variational bottleneck between two layers in the network. Given an input $x$, PRECODE first encodes the input into a representation $z = E(x)$, then samples bottleneck features $b \sim q(b|z)$ from a latent Gaussian distribution. Then, they compute new latent representation $\hat{z} = D(b)$, and finally obtain the output $\hat{y} = O(\hat{z})$. For this defense we focus on an MLP network, which Scheliga et al. (2021) evaluates against several attacks and shows that images cannot be recovered.

Our attack is based on the attack presented in Phong et al. (2017), later extended by Geiping et al. (2020) – it shows that in most cases the inputs to an MLP can be perfectly reconstructed from the gradients. Next we detail this attack as presented in Phong et al. (2017); Geiping et al. (2020). Let the output of first layer of the MLP (before the activation) be $y_0 = A_0 x^T + b_0$, with $x \in \mathbb{R}^{n_0}$,

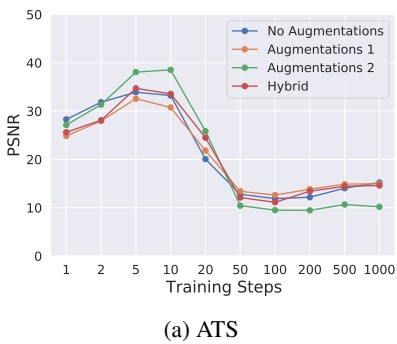

(a) ATS

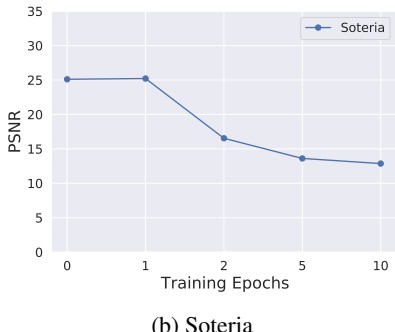

(b) Soteria

Figure 2: PSNR obtained by reconstruction attacks on ATS and Soteria during the first 1000 steps and 10 epochs, respectively, demonstrating high gradient leakage early in training.

$y_0, b_0 \in \mathbb{R}^{n_1}$ and $A_0 \in \mathbb{R}^{n_0 \times n_1}$. Assume a client sends the unperturbed gradients $\frac{dl}{dA_0}$ and $\frac{dl}{db_0}$ for that input layer. Let $(y_0)_i$ and $(b_0)_i$ denote the $i^{\text{th}}$ entry in these vectors. By the chain rule we have:

$$\frac{dl}{d(b_0)_i} = \frac{dl}{d(y_0)_i} \cdot \frac{d(y_0)_i}{d(b_0)_i} = \frac{dl}{d(y_0)_i}.$$

Combining this result with the chain rule for $\frac{dl}{d(A_0)_{i,:}}$ we further obtain:

$$\frac{dl}{d(A_0)_{i,:}} = \frac{dl}{d(y_0)_i} \cdot \frac{d(y_0)_i}{d(A_0)_{i,:}} = \frac{dl}{d(y_0)_i} \cdot x^T = \frac{dl}{d(b_0)_i} \cdot x^T,$$

where $(A_0)_{i,:}$ denotes the $i$-th line of the matrix $A_0$. Assuming $\frac{dl}{d(b_0)_i} \neq 0$, we can precisely calculate $x^T = \left(\frac{dl}{d(b_0)_i}\right)^{-1} \frac{dl}{d(A_0)_{i,:}}$ as there is typically at least one non-zero entry in $b_0$.

## 6 EXPERIMENTAL EVALUATION

We now evaluate existing defenses against strong approximations of the Bayes optimal adversary.

**Evaluating existing defenses** In this experiment, we evaluate the three recently proposed defenses described in Section 5, Soteria, ATS and PRECODE, on the CIFAR-10 dataset (Krizhevsky, 2009). For ATS and Soteria, we use the code from their respective papers. In particular, we evaluated ATS on their ConvNet implementation, a network with 7 convolutional layers, batch-norm, and ReLU activations followed by a single linear layer. We consider 2 different augmentation strategies for ATS, as well as a hybrid strategy. For Soteria, we evaluate our own network architecture with 2 convolutional and 3 linear layers. The defense is applied on the largest linear layer, directly after the convolutions. Both attacks are implemented using the code from Geiping et al. (2020), and using cosine similarity between gradients, the Adam optimizer (Kingma & Ba, 2015) with a learning rate of 0.1, a total variation regularization of $10^{-5}$ for ATS and $4 \times 10^{-4}$ for Soteria, as well as 2000 and 4000 attack iterations respectively. We perform the attack on both networks using batch size 1. For PRECODE, which has no public code, we use our own implementation. We use MLP architecture consisting of 5 layers with 500 neurons each and ReLU activations. Following Scheliga et al. (2021) we apply the variational bottleneck before the last fully connected layer of the network.

The results of this experiment are shown in Fig. 2. We attack ATS for the first 1000 steps of training and Soteria for the first 10 epochs, measuring peak signal-to-noise ratio (PSNR) of the reconstruction obtained using our attack at every step. In all of our PRECODE experiments we obtain perfect reconstruction with PSNR values $> 150$ so we do not show PRECODE on the plot. We can observe that, generally, each network becomes less susceptible to the attack with the increased number of training steps. However, early in training, networks are very vulnerable, and images can be reconstructed almost perfectly. Fig. 3 visualizes the first 40 reconstructed images obtained using our attacks on Soteria, ATS and PRECODE. We can see that for all defenses, our reconstructions are very close to their respective inputs. This indicates that proposed defenses do not reliably protect privacy under gradient leakage, especially in the earlier stages of training. Our findings suggest that creating effective defenses and properly evaluating them remains a key challenge.

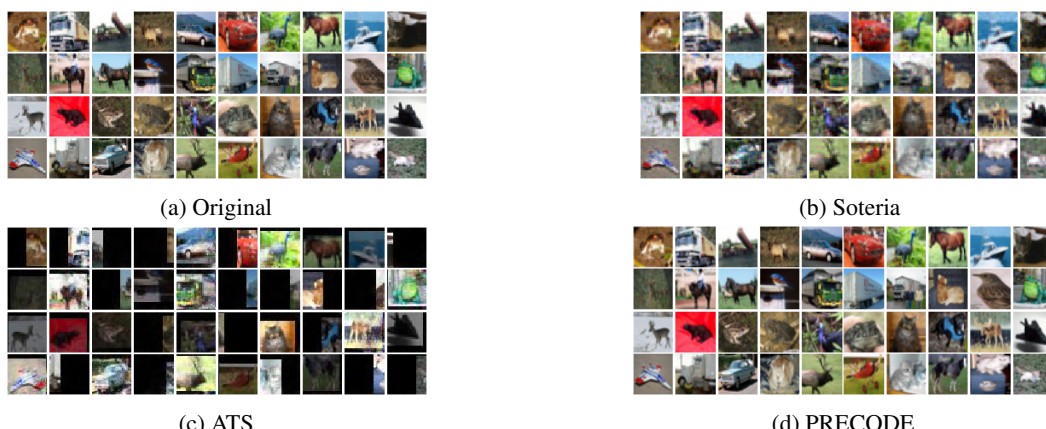

(a) Original                                          (b) Soteria

(c) ATS                                               (d) PRECODE

Figure 3: Images obtained by running attacks on Soteria, ATS, and PRECODE on the CIFAR-10 dataset after the $10^{\text{th}}$ training step. We can observe that reconstructed images are very close to the original ones, meaning that these defenses do not protect privacy early in the training.

Table 2: PSNR of reconstructed images using approximate Bayes optimal attack, and attacks based on $\ell_2$, $\ell_1$, and cosine distance between the gradients. We confirm our theoretical results that Bayes optimal attack performs significantly better when it knows probability distribution of the gradients.

|  | | Train step 0 | | | | Train step 500 | | | |
| --- | --- | --- | --- | --- | --- | --- | --- | --- | --- |
|  | Defense | Bayes | $\ell_2$ | $\ell_1$ | Cos | Bayes | $\ell_2$ | $\ell_1$ | Cos |
| MNIST | Gaussian | 19.63 | **19.95** | 19.39 | 19.70 | 16.06 | **16.18** | 15.86 | 15.39 |
|  | Laplacian | 19.48 | 18.86 | **19.99** | 18.64 | 15.95 | 15.47 | **16.31** | 15.14 |
|  | Prune + Gauss | **18.40** | 14.44 | 13.95 | 16.72 | **16.42** | 13.15 | 13.24 | 14.47 |
|  | Prune + Lap | **18.27** | 14.01 | 13.86 | 15.21 | **16.52** | 13.60 | 13.54 | 14.57 |
| CIFAR-10 | Gaussian | **21.86** | 21.81 | 21.49 | 21.53 | 17.94 | **18.64** | 18.00 | 18.34 |
|  | Laplacian | 21.87 | 21.02 | **21.89** | 20.87 | 18.41 | 18.37 | **19.11** | 18.02 |
|  | Prune + Gauss | **20.73** | 16.60 | 16.20 | 18.73 | **18.67** | 15.61 | 15.27 | 16.74 |
|  | Prune + Lap | **20.54** | 16.25 | 16.13 | 17.40 | **18.51** | 15.32 | 15.24 | 15.51 |

**Bayes optimal adversary in practice** In the following experiment, we compare the approximation of Bayes optimal adversary described in Section 4 with three variants of Inverting Gradients attack (Geiping et al., 2020), based on $\ell_2$, $\ell_1$, and cosine distance, on the MNIST (Lecun et al., 1998) and CIFAR-10 datasets. Recall from Table 1 that the $\ell_2$, $\ell_1$ and cosine attacks make different assumptions on the probability distribution of the gradients, which are not optimal for every defense. To this end, we evaluate the performance of these attacks on a diverse set of defenses. For both MNIST and CIFAR-10, we train a CNN with ReLU activations, and attack gradients created on image batches of size 1 both at the initialization and step 500 of the training. For all attacks, we use anisotropic total variation image prior, and we initialize the images with random Gaussian noise. Further experiments using prior based on pixel value range are presented in Appendix A.4. We optimize the loss using Adam (Kingma & Ba, 2015) with exponential learning rate decay. For this experiment, we modified Eq. (3) to add an additional weighting parameter $\beta$ for the prior as $\frac{1}{k} \sum_{i=1}^{k} \log p(g|x_i) + \beta \log p(x_i)$ which compensates for the imperfect image prior selected.

Experimentally, we observed that the success of the different attacks is very sensitive to the choice of their parameters. Therefore, to accurately compare the attacks, we use grid search that selects the optimal parameters for each of them individually. In particular, for all attacks we tune their initial learning rates and learning rate decay factors as well as the weighting parameter $\beta$. Further, for the $\ell_2$, $\ell_1$ and cosine attacks the grid search also selects whether exponential layer weighting, introduced by Geiping et al. (2020), is applied on the gradient reconstruction loss. In this experiment, the approximation of the Bayes optimal adversary uses $k = 1$ Monte Carlo samples. Further ablation study on the effects of $k$ is provided in Appendix A.6. Our grid search explores 480 combinations of parameters for the $\ell_2$, $\ell_1$ and cosine attacks, and 540 combinations for the Bayesian attacks, respectively. We provide more details on this parameter search in Appendix A.3.

For this experiment, we consider defenses for which we can exactly compute conditional probability distribution of the gradients for a given input, denoted earlier as $p(g|x)$. First we consider two defenses which add Gaussian noise with standard deviation $\sigma = 0.1$ and Laplacian noise of scale $b = 0.1$ to the original gradients, thus corresponding to the probability distributions $p(g|x) = \mathcal{N}(g - \nabla_\theta l(h_\theta(x), y), \sigma^2 I)$ and $p(g|x) = Lap(g - \nabla_\theta l(h_\theta(x), y), bI)$, respectively. Note that the Gaussian defense with additional gradient clipping corresponds to the well known DP-SGD algorithm (Abadi et al., 2016). We also consider defenses based on soft pruning that set random 50% entries of the gradient to 0, and then add Gaussian or Laplacian noise as in previous defenses. In Table 2 we report the average PSNR values on the first 100 images of the training set for each combination of attack and defense. First, we observe, as in Fig. 2, that network is significantly more vulnerable early in training. We can observe that Bayes optimal adversary generally performs best, showing that the optimal attack needs to leverage structure of the probability distribution of the gradients induced by the defense. Note that, in the case of Gaussian defense, $\ell_2$ and Bayes attacks are equivalent up to a constant factor (as explained in Section 4), and it is expected that they achieve a similar result. In all other cases, Bayes optimal adversary outperforms the other two attacks. Overall, this experiment provides empirical evidence for our theoretical results from Section 4.

**Approximations of Bayes optimal adversary** In this experiment, we compare the Bayes optimal adversary with attacks that have suboptimal approximations of $p(g|x)$ and $p(x)$. As vision datasets have complex priors $p(x)$ which we cannot compute exactly, we now consider a synthetic dataset where $p(x)$ is 20-dimensional unit Gaussian. We define the true label $y := \arg\max(Wx)$ where $W$ is a fixed random matrix, and perform an attack on a 2-layer MLP defended by adding Laplacian noise with a 0.1 scale. Thus, $p(x)$ is a Gaussian, and $p(g|x)$ is Laplacian. In this study, we consider 4 different variants

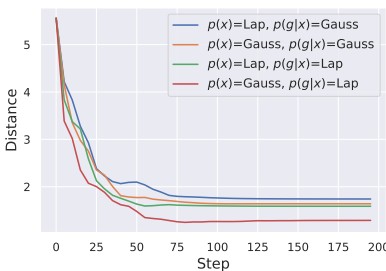

Figure 4: Ablation with the Bayes attack.

of the attack, obtained by choosing Laplacian or Gaussian for prior and conditional. Fig. 4 shows, for each attack, the distance from the original input for 200 steps. We can observe that Bayes optimal attack (with Gaussian prior and Laplacian conditional) converges significantly closer to the original input than the other attacks, providing empirical evidence for our theoretical results in Section 4.

**Discussion and future work** Our experimental evaluation suggests the evaluation of the existing heuristic defenses against gradient leakage is inadequate. As these defenses are significantly more vulnerable at the beginning of the training, we advocate for their evaluation throughout the *entire* training, and not only at the end. Ideally, defenses should be evaluated against the Bayes optimal adversary, and if such adversary cannot be computed, against properly tuned approximations such as the ones outlined in Section 4. Some interesting future work includes designing better approximations of the Bayes optimal adversary, e.g. by using better priors than total variation, and designing an effective defense for which it is tractable to compute the distribution $p(g|x)$ so that it can be evaluated using Bayes optimal adversary similarly to what we did in Table 2. We believe our findings can facilitate future progress in this area, both in terms of attacks and defenses.

## 7 CONCLUSION

We proposed a theoretical framework to formally analyze the problem of gradient leakage, which has recently emerged as an important privacy issue for federated learning. Our framework enables us to analyze the Bayes optimal adversary for this setting and phrase it as an optimization problem. We interpreted several previously proposed attacks as approximations of the Bayes optimal adversary, each approximation implicitly using different assumptions on the distribution over inputs and gradients. Our experimental evaluation shows that the Bayes optimal adversary is effective in practical scenarios in which it knows the underlying distributions. We additionally experimented with several proposed defenses based on heuristics and found that they do not offer effective protection against stronger attacks. Given our findings, we believe that formulating an effective defense that balances accuracy and protection against gradient leakage during all stages of training remains an exciting open challenge.

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

| Conv2d(in_channels=3, out_channels=32, kernel_size=3, stride=1, padding=1) |
|:---:|
| ReLU() |
| AvgPool2d(kernel_size=2, stride=2) |
| Conv2d(in_channels=32, out_channels=64, kernel_size=1, padding=1) |
| ReLU() |
| AvgPool2d(kernel_size=2, stride=2) |
| Linear(in_features=5184, out_features=2000) |
| ReLU() |
| Linear(in_features=2000, out_features=1000) |
| ReLU() |
| Linear(in_features=1000, out_features=10) |

Table 3: ConvBig architecture.

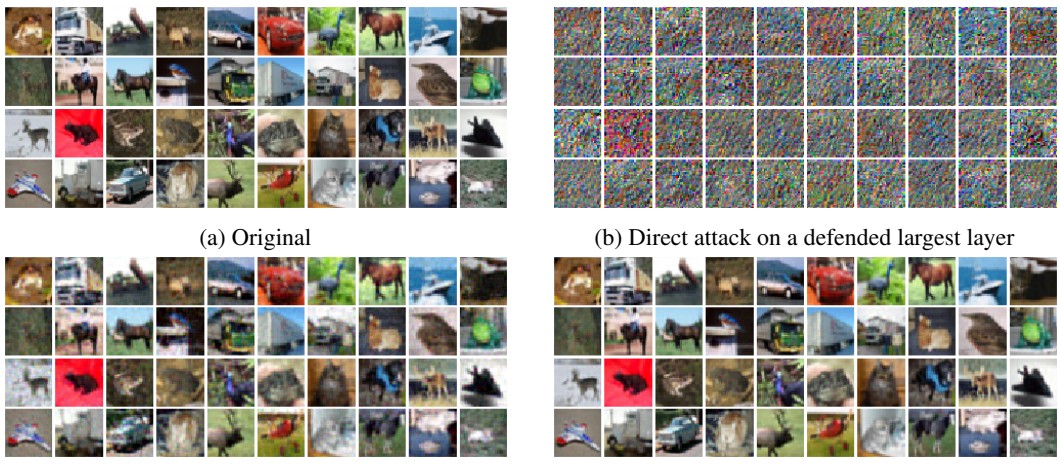

(a) Original

(b) Direct attack on a defended largest layer

(c) Our attack on the defended largest layer

(d) Our attack on a defended second largest layer

Figure 5: We compare the results of a direct attack on the defended network 5b with our attack 5c. Furthermore we show our attack on a network in which the second largest layer is defended 5d. All networks have been trained for 10 steps.

# A  APPENDIX

Here we provide additional details for our work.

## A.1  SOTERIA

For Soteria, we built directly on the Sun et al. (2021) repository using their implementation of the defense and the included Inverting Gradient (Geiping et al., 2020) library. Testing was conducted primarily on the ConvBig architecture presented below:

The architecture consists of a convolutional "feature extractor" followed by three linear layers, shown in Table 3. We chose this architecture because (i) it is simple in structure while providing reasonable accuracies on datasets such as CIFAR10 and (ii) because (unlike many small convolutional networks) it has more than one layer with a significant fraction of the overall network parameters. In particular, the first linear layer roughly contains around $80\%$ of the network weights and the second one $20\%$. This is particularly relevant for our attack, as cutting out a large layer of weights will negatively affect the reconstruction quality. As it is in practice uncommon to have the majority of weights in a single layer, we believe our architecture provides a reasonable abstraction. To justify this, we show in Fig. 5 that no matter which of the larger two layers is defended by Soteria, we can attack the network successfully. For all Soteria defenses we set the pruning rate (refer to (Sun et al., 2021) for details) to $80\%$. Note however that our attack works independent of the pruning rate, as we always remove the entire layer.

| |
|---|
| Conv2d(3, 1 * width, kernel_size=3, padding=1), BatchNorm2d(), ReLU() |
| Conv2d(1 * width, 2 * width, kernel_size=3, padding=1), BatchNorm2d(), ReLU() |
| Conv2d(2 * width, 2 * width, kernel_size=3, padding=1), BatchNorm2d(), ReLU() |
| Conv2d(2 * width, 4 * width, kernel_size=3, padding=1), BatchNorm2d(), ReLU() |
| Conv2d(4 * width, 4 * width, kernel_size=3, padding=1), BatchNorm2d(), ReLU() |
| Conv2d(4 * width, 4 * width, kernel_size=3, padding=1), BatchNorm2d(), ReLU() |
| MaxPool2d(3), |
| Conv2d(4 * width, 4 * width, kernel_size=3, padding=1), BatchNorm2d(), ReLU() |
| Conv2d(4 * width, 4 * width, kernel_size=3, padding=1), BatchNorm2d(), ReLU() |
| Conv2d(4 * width, 4 * width, kernel_size=3, padding=1), BatchNorm2d(), ReLU() |
| MaxPool2d(3) |
| Linear(36 * width, 3) |

Table 4: ConvNet architecture. Our benchmark instatiation uses a width of $64$.

We use the Adam optimizer with a learning rate of 0.1 with decay. As similarity measure, we use cosine similarity. Besides cutting one layer of weights, we weigh all gradients equally. The total variation regularization constant is $4 \times 10^{-4}$. We initialize the initial guess randomly and only try reconstruction once per image, attacking one image at a time. For training, we used a batch size of 32. We trained the network without defense and applied the defense at inference time to speed up the training.

## A.2 AUTOMATED TRANSFORMATION SEARCH

For ATS, we built upon the repository released alongside (Gao et al., 2021). We use the ConvNet architecture with a width of $64$ also proposed in (Gao et al., 2021) and train with the augmentations "7-4-15", "21-13-3", "21-13-3+7-4-15" which perform the best on ConvNet with CIFAR100. We present the ConvNet architecture in Table 4.

For reconstruction, we use the Adam optimizer with a learning rate of 0.1 with decay. As similarity measure, we use cosine similarity weighing all gradients equally. The total variation regularization constant is $1 \times 10^{-5}$. We initialize the initial guess randomly and only try reconstruction once per image, attacking one image at a time. For training, we used a batch size of 32 and trained individually for every set of augmentations.

In Figure 6 we show how the quality of the reconstructed inputs degrades during training. Nevertheless we can recover high-quality inputs during the first 10 to 20 training steps.

## A.3 PARAMETER SEARCH FOR THE ATTACKS

Since the range of $\beta$ for which the different attacks perform well is wide, prior to the grid search we need to calculate a range of reasonable values for $\beta$ for each of the attacks. We do this by searching for values of $\beta$ for which the respective attack is optimal. The rest of the parameters of these attacks are set to the same set of initial values. The values of $\beta$ considered are in the range $[1 \times 10^{-7}, 1 \times 10^{5}]$ and are tested on logarithmic scale. The final range for $\beta$ used in the grid search is given by the range $[0.5\beta_*, 2\beta_*]$, where $\beta_*$ is the values that produced the highest PSNR.

## A.4 EXPERIMENT WITH STRONGER PRIORS

In this section, we compare the effects of different image priors on the gradient leakage attack. In particular, we compared a simple anisotropic total variation image prior and a prior that uses combination of the same anisotropic total variation and an error term first introduced in Geng et al. (2021) that encourages the reconstructed image's pixels to be in the $[0, 1]$ range:

$$\log p(x) = \phi \cdot L_{\text{TV}} + (1 - \phi) \cdot L_{[0,1]}$$
$$L_{[0,1]} = \|x - \text{clip}(x, 0, 1)\|_2,$$

where $\phi \in [0, 1]$ balances the total variation error term $L_{\text{TV}}$ and the pixel range error term $L_{[0,1]}$ and $\text{clip}(x, 0, 1)$ clips the values of $x$ in the $[0, 1]$ range. We compared both priors on the MNIST

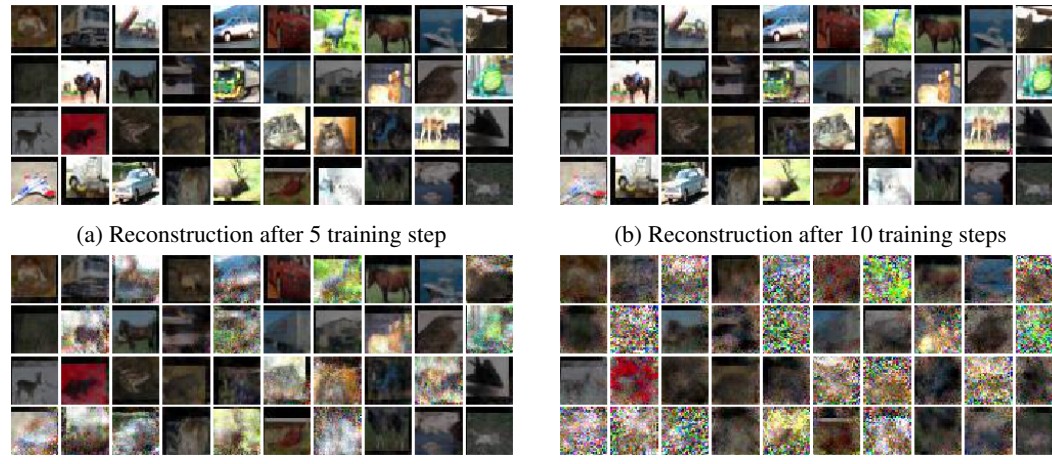

(a) Reconstruction after 5 training step          (b) Reconstruction after 10 training steps

(c) Reconstruction after 20 training steps          (d) Reconstruction after 50 training steps

Figure 6: Our reconstruction results after several training steps with ATS, batch_size 32, and augmentations 7-4-15. We can see how the visual quality starts to decline after 10 steps and at 50 steps one can no longer reliably recover the input.

Table 5: Attack using $l_2$ distance metric on MNIST trained for 500 steps with and without using stronger prior that takes into account that pixels should be in the $[0, 1]$ range.

| Defense | $\ell_2 + L_{\text{TV}}$ | $\ell_2 + L_{\text{TV}} + L_{[0,1]}$ |
|---|---|---|
| Gaussian | 16.18 | **16.30** |
| Laplacian | 15.47 | **15.76** |
| Prune + Gauss | 13.15 | **13.63** |
| Prune + Lap | 13.60 | **14.10** |

network trained for 500 steps from Table 2 using $\ell_2$ conditional distribution. We compared on all defenses considered in Table 2. We used the same grid search procedure, but we extended it to search for optimal value of $\phi$ as well. The results are presented in Table 5. From the results, we see that the addition of the pixel range prior term improved the attack for all defenses, showing that despite its simplicity the pixel range prior is very effective.

A.5    OUR PRACTICAL ATTACKS AS APPROXIMATIONS OF THE BAYES OPTIMAL ADVERSARY

In this subsection we interpret our attacks on Soteria, ATS and PRECODE as different approximations of Bayes optimal adversary. We summarize this in Table 6, and also describe details of each approximation in separate paragraphs.

**Attack on Soteria**    In this section, we interpret the attack on Soteria that we presented in Section 5, as an approximation to the Bayes optimal attack in this setting. As described in Section 5, Soteria takes the original network gradient $\nabla W = \{\nabla W_1, \nabla W_2, \ldots, \nabla W_L\}$ and defends it by substituting the gradient at layer $l$ with the modified gradient $\nabla W_l'$, producing

$$g_k = \begin{cases} \nabla W_m & \text{if } m \neq l \\ \nabla W_l' & \text{if } k = l \end{cases},$$

where $g_m$ is the subvector of the client gradient update vector $g$, corresponding to the $m^{\text{th}}$ layer of the network.

In Section 4, we derived the Bayes optimal attacker objective as

$$\frac{1}{k}\sum_{i=1}^{k}(\log p(g|x_i) + \beta \log p(x_i)).$$

| Attack | Prior $p(x)$ | Conditional $p(g\|x)$ |
|---|---|---|
| Soteria attack | TV | Layerwise Gaussian |
| ATS attack | TV | Cosine |
| PRECODE attack | Arbitrary | Dirac delta mixture |

Table 6: Our attacks on heuristic defenses can be interpreted as instances of our Bayesian framework. We show prior and conditional distribution for corresponding losses that each attack uses.

Next, we show how we approximate this objective for the Soteria defense. Since Soteria modifies the network gradients per-layer, we model the gradient probability distribution $p(g|x)$ of our Bayes optimal attack as per-layer separable:

$$p(g|x) = \prod_{m=1}^{L} p(g_m|x).$$

To model the design choice of Soteria to not change gradients of layers different than $l$, we set $p(g_m|x) = \mathcal{N}(\nabla W_m, \sigma_m)$ with $\sigma_m \to 0$, for all layers $m \neq l$. With that choice, the Bayes objective becomes

$$\frac{1}{k} \sum_{i=1}^{k} (\log p(g|x_i) + \beta \log p(x_i))$$

$$= \frac{1}{k} \sum_{i=1}^{k} (\log p(g_l|x_i) + \beta \log p(x_i) + \sum_{m \neq l} \log p(g_m|x_i)) \tag{5}$$

$$= C + \frac{1}{k} \sum_{i=1}^{k} (\log p(g_l|x_i) + \beta \log p(x_i) - \frac{1}{2} \sum_{m \neq l} \frac{1}{\sigma_m^2} ||g_m - \nabla_\theta l(h_\theta(x_i), y)||_2^2).$$

As $\sigma_m \to 0$, the contribution of $\log p(g_l|x)$ to the overall objective becomes exceedingly smaller. Therefore, one can choose $\sigma_m = \sigma$ for some $\sigma > 0$, such that the optimal solution of the Bayes objective in Eq. (5) is very close to the optimal solution of the surrogate objective:

$$\frac{1}{k} \sum_{i=1}^{k} (\beta \log p(x_i) - \frac{1}{2} \sum_{m \neq l} \frac{1}{\sigma^2} ||g_m - \nabla_\theta l(h_\theta(x_i), y)||_2^2). \tag{6}$$

The objective in Eq. (6) is the exact objective that we optimize to produce the Soteria attacks in Section 5.

**Attack on ATS**   As described in Section 5, we attack ATS using the attack from Geiping et al. (2020). This attack corresponds to the prior with total variation, and conditional distribution is based on cosine similarity.

**Attack on PRECODE**   Our attack on PRECODE is based on the observation that the distribution $p(x|g)$ for PRECODE corresponds to a Dirac delta distribution. Namely, given gradient $g$, there is always unique input $x_0$ that produced this gradient, and the density $p(x|g)$ is then a Dirac delta centered at $x_0$ (uniqueness of the solution follows from the derivation in Phong et al. (2017) for linear layers). As shown in Eq. (2), this $x_0$ corresponds to maximizing the value of $p(x|g)$. In this case, we can in fact solve for the maximum of $p(x|g)$ exactly using attack described in Section 5, and we do not have to rewrite the objective using Bayes rule and apply optimization. In terms of $p(x)$ and $p(g|x)$, we can have arbitrary $p(x)$, while $p(g|x)$ is mixture of Dirac delta distributions where the mixture comes from the random samples of the VAE used by PRECODE.

### A.6   ABLATION STUDY OF THE MONTE CARLO ESTIMATION

In this experiment we perform an ablation study for the Monte Carlo estimation of the adversarial objective. More specifically, we vary the number of samples $k$ used for the Monte Carlo estimate.

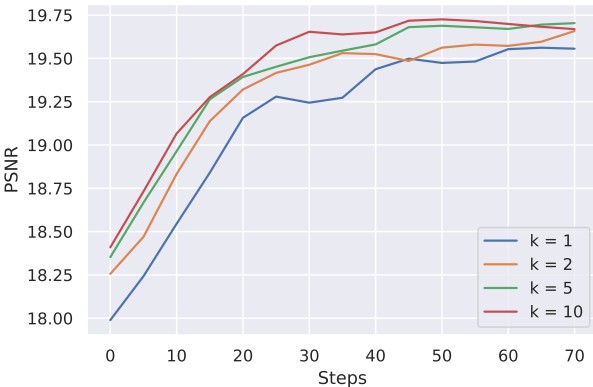

Figure 7: Ablation for the number of samples in the Monte Carlo estimate.

We show the PSNR value averaged over 100 different sampled images for each step of the optimization process in Figure 7. We used a convolutional network on the MNIST dataset, at the initial training step, defended by Gaussian noise of standard deviation 0.1 and using $\delta = 9.0$ in the definition of the adversarial risk. The results indicate that higher number of samples in the Monte Carlo estimate results in faster convergence of the attacker towards the reconstruction image closer to the original. This is expected as estimate using larger number of samples results in an Monte Carlo estimator that has lower variance.

## A.7 RELATIONSHIP BETWEEN BAYES ATTACK AND LANGEVIN DYNAMICS

In this section, we discuss the relationship between our proposed Bayes optimal attack and Langevin updates proposed by Yin et al. (2021). We focus on the case where the number of samples in the Monte Carlo estimate is $k = 1$. As shown in Algorithm 1, our update is

$$x \leftarrow x + \alpha \nabla_x (\log p(g|x_1) + \log p(x_1)),$$

where $x_1$ is a point sampled from the ball $B(x, \delta)$ uniformly at random. At the same time, the update with Langevin dynamics proposed by Yin et al. (2021) is given by

$$x \leftarrow x + \alpha \left( \alpha_n \eta + \nabla_x (\log p(g|x) + \log p(x)) \right),$$

where $\eta \sim \mathcal{N}(0, \mathcal{I})$ is noise sampled from unit Gaussian, and $\alpha_n$ is noise scaling coefficient. As Yin et al. (2021) explain, we can view Langevin updates as a method to encourage exploration and diversity. The main differences between the updates are: (i) we sample a point $x_1$ from the ball $B(x, \delta)$ uniformly at random, while Langevin update in Yin et al. (2021) samples noise $\eta$ from unit Gaussian, (ii) we compute gradient at the sampled point $x_1 \sim B(x, \delta)$, while Langevin update evaluates gradient directly at $x$, and adds noise $\eta$ to $x$ afterwards. Note that Yin et al. (2021) do not provide ablation study to investigate the effect of Langevin update, so we do not know important it is in the overall optimization. Overall, both Monte Carlo estimation and Langevin updates can be seen as different ways to add noise to the optimization process, though currently we do not see a way to formally view Langevin updates as a Bayesian prior.

