# OpenReview forum: "Bayesian Framework for Gradient Leakage"
_ICLR.cc/2022/Conference — ICLR 2022 Poster_

### Official Review · Reviewer_cPAG · 2021-11-01

**Correctness:** 3
**Technical Novelty And Significance:** 3
**Empirical Novelty And Significance:** 1
**Recommendation:** 6
**Confidence:** 4

**Main Review:**

+ This paper is well-written, easy to follow. Provides sufficient background and literature review.
+ The framework is straightforward yet makes sense. The mathematical results seem to be derived correctly.
+ Linking the existing attacks to their framework is interesting.
-  As mentioned in the paper the Bayesian adversary loss is intractable, thus providing some kind of approximation for it would have been better.
- It is not clear to me even though as mentioned by the authors we do not have any sort of effective defense, what are the authors' suggestions in that regard as the paper is based on that. Could the authors please elaborate on that?
- Section 6, the existing attack section contains interesting results but I fail to see the relevance to this paper. It has taken most of the section without justifying its importance to the proposed framework.
- This paper suffers from the lack of a wide range of experimental results to back up the claims. The experimental results must cover compassion to other techniques in addition to accenting the importance of each experiment.
-  The lack of provided details in the paper, however, the code for experimental results must not be considered due to the late submission of their codes. Authors should provide more details regarding their experiments.



**Summary Of The Paper:**

This paper introduces a framework to address issues in the leakage of information through gradients. This paper claims to provide Bayes optimal adversary as an optimization problem which other attacks can be considered as an approximation of this framework. Some experiments are provided to support their claims.

**Summary Of The Review:**

- Even though the assumptions seem to be correct, the contribution of the paper is extermley simple and straightforward.
- The experimental results seem to be insufficient as it is only compared to two other methods. The experiment section is written poorly and the importance of each section is not justified.

---

> ### Author Response · Authors · 2021-11-14
> **Response to Reviewer cPAG**
>
> Thank you for your feedback and suggestions. We address your questions below, and we also provide a response to the common question raised by several reviewers in a separate comment.
>
> **Could you provide some approximation of the intractable Bayesian adversary loss?**
>
> Yes, in fact all attacks listed in Table 1 are different approximations of the loss used by Bayes optimal adversary. For more detailed discussion, please see the last paragraph in Section 4 where we derive the approximations used in this paper.
>
> **Could authors suggest what defense should be used in practice, based on their findings?**
>
> Yes, our work suggests that defenses based on heuristics do not work, as the adversary can still reconstruct the original images during early epochs of training. Thus, defenses based on differential privacy (DP) remain the only viable option that truly protects privacy of the user data during training. However, the main drawback is that models trained with DP have low accuracy, and thus it remains an open problem to create a defense that would effectively balance accuracy and privacy.
>
> **This paper suffers from the lack of a wide range of experimental results to back up the claims. The experimental results must cover compassion to other techniques in addition to accenting the importance of each experiment.**
>
> Note that our experiments are quite comprehensive:
> - We tested 3 existing defenses (Soteria, ATS, PRECODE)
> - We compared Bayes optimal adversary against 2 different attacks from prior work (and we now added 1 more that was suggested by the other reviewer), on 2 different datasets (MNIST, CIFAR-10), at 2 different training checkpoints (0 and 500 steps), with separate tuning of best parameters for each attack, as described in Appendix A.2
>
> We are open to suggestions for additional experiments that the reviewer might have in mind.
>
> The key takeaways from our experiments are:
> 1. Existing defenses based on heuristics do not effectively protect against strong adversaries at the beginning of training.
> 2. Bayes optimal adversary can leverage knowledge of prior $p(x)$ and conditional $p(g|x)$ to perform better than the attacks that do not take these distributions into account.
>
> **Could you provide more experimental details?**
>
> We provide all details necessary to reproduce the experiments (note that some details are in the Appendix due to the lack of space), but if the reviewer can provide us with some concrete detail that is missing, we would be glad to include it into the paper.
>
> **The code must not be considered due to the late submission.**
>
> We respectfully disagree and we believe our code should indeed be considered. Here we cite directly from the FAQ on Author Guide (https://iclr.cc/Conferences/2022/AuthorGuide):
>
> *Q: How can we make our code available for reviewing anonymously?
> You can share your code in three ways:*
> 1. *Anonymize your code, put it in a .zip file and submit it as supplementary materials.*
> 2. *Make an anonymous repository and put the link in your paper.*
> 3. *The above methods will make your code public, along with your paper and reviews/comments for the paper.*
> *After we open the discussion forums for all submitted papers, make a comment directed to the reviewers and area chairs and put a link to an anonymous repository.*
>
> Thus, we believe that our code submission follows the 3rd point of the ICLR guidelines above.

---

> > ### Comment · Reviewer_cPAG · 2021-11-22
> > **Response**
> >
> > Thank you for clarifications.
> >
> > -- Could you provide some approximation of the intractable Bayesian adversary loss?
> >
> > While the attacks mentioned in section 4 can be considered empirical approximations of the Bayesian adversary loss, some mathematical approximation is required to make the framework more practical. At this point the proposed framework is not practical as also pointed out by Reviewer DWMa
> >
> > -- Could authors suggest what defense should be used in practice, based on their findings?
> >
> > I appreciate the authors' suggestions. However, the claim that networks are more vulnerable during early stages has been already shown by earlier works (as pointed out by Reviewer DWMa) and in my opinion, cannot be considered a finding.
> > Could you provide more experimental details?
> >
> > I understand that due to lack of space the experimental setups are deferred to the supplementary material, but lack of no experimental settings significantly hurts the reproducibility of the results and the clarity of the paper. The authors should consider moving important settings to the main paper.
> >
> > I'd also like to point out that the code was not to consider for the initial evaluation and taken into account.
> >
> > After reading author's response, I will keep my score.

---

> > > ### Author Response · Authors · 2021-11-23
> > > **Response**
> > >
> > > Thank you for the response. We address your concerns below.
> > >
> > > **Is the framework not practical without some kind of mathematical approximation?**
> > >
> > > Note that attacks in Section 4 are obtained using a mathematical approximation: we first apply Jensen’s inequality, and then estimate the integral using Monte Carlo sampling. Our experiments in Section 6 show that these approximations result in practical attacks that perform better than the other attacks that do not take the probability distribution into the account. Furthermore, note that we have now added Appendix A.6 which shows that our attacks on existing defenses can be viewed as mathematical approximations of Bayes optimal attack.
> > > It would be great if the reviewer could clarify more precisely what they mean by mathematical approximation, so we could better address the concerns.
> > >
> > > **Could the claim that networks are more vulnerable during early stages be considered a finding?**
> > >
> > > While it is correct that prior work (e.g. Geiping et al.) has already shown that *undefended* networks are easier to attack at the early stages, as we wrote to Review DWMa, our contribution is showing that the problem also occurs for *defended* networks, as the evaluation of the early stages is missing from the papers that introduced the respective defenses. This is the key limitation of proposed defenses, and it is not natural to expect that this limitation holds for any defense mechanism (given that DP does not have this problem). Thus, the key takeaway from our evaluation is that heuristic defenses do not offer any privacy protection (unlike DP which offers protection throughout training), and users should avoid building systems based on these particular defenses, which was not clear before our paper.
> > >
> > > **Could you provide more experimental details? Would you consider moving important settings to the main section of the paper?**
> > >
> > > Yes, we will move more of the important experimental setup details to the main body in the next revision of the paper. We believe that all of the details necessary to reproduce our work are present either in Section 6 or in the Appendix. If the reviewer could further clarify which experimental details are missing, it would help us in making a revision.

---

### Official Review · Reviewer_CRtt · 2021-11-02

**Correctness:** 4
**Technical Novelty And Significance:** 3
**Empirical Novelty And Significance:** 2
**Recommendation:** 6
**Confidence:** 2

**Main Review:**

After reading author's response: review remains unaltered.

Strengths:
1. The unification of several existing attacks is interesting.
2. The paper is well-written.


Weaknesses:
1. It would be better if the authors could introduce more details of how to approximate p(x) and p(g|x). Estimating it from data is known to be a hard problem. Does this mean the attack may fail on a large-scale dataset?


**Summary Of The Paper:**

This paper provides a gradient leakage attack via Bayes optimal adversary. The authors also demonstrate that existing attacks can be seen as approximations of Bayes optimal adversary. Empirically, this paper shows some heuristic defenses fail under the proposed attack.

**Summary Of The Review:**

This paper is overall well-written. The proposed method is simple and empirically effective.

---

> ### Author Response · Authors · 2021-11-14
> **Response to Reviewer CRtt**
>
> Thank you for your feedback and suggestions. We address your questions below, and we also provide a response to the common question raised by several reviewers in a separate comment.
>
> **Could you provide more details how to estimate $p(x)$ and $p(g|x)$? Could the attack fail on a large-scale dataset?**
>
> Note that prior work has proposed different ways to estimate $p(x)$ and $p(g|x)$, summarized in Table 1. More detailed description can be found in the last paragraph of Section 4 where we explain for each proposed attack to which distribution it corresponds to. Even though these attacks are only approximations, they perform well in practice, even for large-scale datasets: e.g. Yin et al. demonstrated that they can reconstruct ImageNet images for large batch sizes.

---

### Official Review · Reviewer_6AUa · 2021-11-02

**Correctness:** 4
**Technical Novelty And Significance:** 3
**Empirical Novelty And Significance:** 3
**Recommendation:** 8
**Confidence:** 5

**Main Review:**

In summary I think this is an interesting submission. The unification of existing attacks in a Bayesian sense (while maybe not thrilling from a purely theoretical point of view) is certainly a good idea and clearly described and executed and, to me, the strongest part of this work. The experiments in Sec. 6 then show that this viewpoint can have tangible benefits. However I am torn regarding the section on breaking existing defenses. While these are great attacks and I liked to read this section, it does feel disconnected from the rest of the paper. None of the insights from the Bayesian framework seem to help in creating these adaptive attacks.

Some more comments and questions regarding the experimental evaluation below:

* After the introduction of the Bayesian framework in Sec. 4 it does seem a bit disappointing that the only Bayesian attack evaluated in Sec. 6 operates with k=1 Monte-Carlo samples. It would have been great to see an ablation of the effects (or the lack thereof) of a range of values of k.
* If k=1 and p(g|x) is Gaussian, how does the proposed attack differ from the attack of Yin et al. with Langevin noise (from a random Gaussian distr.)? The order of gradient computation and noise addition differs, but it is not clear to me if this is a meaningful distinction in the Bayesian sense.
* For Sec. 6, Table 2 it would be interesting to complete the selection of objectives and also show results for an $\ell^1$ objective, which would be the optimal adversary for Laplacian noise when $\delta\to0$.
* The data distribution $p(x)$ appears somewhat under-utilized after its introduction in Sec. 4. For example, for the TV regularization, the optimal $\alpha$ could have been tuned by fitting the distribution to measured data instead of grid-searching all possible alpha in terms of final PSNR of the attack. Also, would the results in Table 2 be strengthened or weakened when using a stronger data prior such as e.g. the  DeepInversion prior from Yin et al.?
* In terms of discussions of priors, Table 1 should also contain a mention of Jeon et al., who do spend time on searching for closer approximations to $p(x)$ in several ways.
* One particular prior effect that appears in some previous work but sometime falls under the radar is the prior knowledge that all real images are drawn from the bounded set  $[0,1]^n$  (for $n$ the number of pixels). Did the authors experiment with this (although arguably relatively weak) prior as well?
* Also in Table 2, the authors also compare to pruning. However it is not clear to me whether $p(g|x)$ takes the pruning into account in those rows?

**Summary Of The Paper:**

The submission "Bayesian Framework for Gradient Leakage" discusses privacy attacks against federated learning based on gradient inversion. In the first part of this work, a range of existing attacks is unified in a Bayesian setting. In the middle part of this work, three existing defenses are analyzed and broken by adaptive attacks. In the final part of this work, an attack based on estimation of the optimal Bayesian adversary is evaluated against several classical defenses and compared to other objectives.

**Summary Of The Review:**

I have a positive outlook on this submission. Classifying and unifying previous attacks in a single framework and clarifying the optimal choice of regularizer and objective is a helpful contribution to the community.

I do have several questions regarding the experimental evaluation that I would like for the authors to answer and I would be interested in a clarification why the inclusion of Sec. 5 should be part of this submission and its connection to the overall Bayesian framework discussed in this work.

--------------------------------------------- After Author Responses: --------------------------------------------

The authors have adressed my concerns during the response period. This lead me to increase my score from 6 to 8.

---

> ### Author Response · Authors · 2021-11-14
> **Response to Reviewer 6AUa**
>
> Thank you for your feedback and suggestions, which we now incorporated into the new revision of the paper. We address your questions below, and we also provide a response to the common question raised by several reviewers in a separate comment.
>
> **Can you provide ablation study for using different $k$ in Monte Carlo estimation?**
>
> Yes, we provide the results of this experiment in Appendix A.4, which we briefly summarize here. We experimented with the number of samples $k = 1, 2, 5, 10$ and measured the PSNR between the reconstruction and the original image at every step of the attack. Our results indicate that using more samples indeed results in faster convergence towards the reconstruction closest to the original image, as expected based on the theory.
>
> **If $k = 1$ and $p(g|x)$ is Gaussian, how does the proposed attack differ from the attack of Yin et al. with Langevin noise?**
>
> Note that Langevin noise in Yin et al. is added to encourage exploration for the optimization process used, and our Bayesian framework is orthogonal to the actual optimization process used to maximize the final approximation of the objective. Investigating the optimization process of the final objective would certainly be an interesting item for future work.
>
> **Can you add an attack with $\ell_1$ objective to Table 2?**
>
> Yes, we added an attack which uses  $\ell_1$  objective to Table 2. The results indicate that  $\ell_1$ indeed works best for the Laplacian defense, confirming our theoretical findings from Section 4. For defenses other than Laplacian, Bayes optimal attacker expectedly performs better.
>
> **Could $\alpha$ of TV be found by fitting to the data distribution instead of grid-search? Could results from Table 2 be strengthened by using stronger prior such as DeepInversion?**
>
> Note that one problem with fitting total variation this way is that we cannot compute the normalization constant for total variation, and thus we need an approach that works with unnormalized probabilities, so we used grid search. The other point of using stronger priors certainly makes sense, and we believe this is an interesting future work item. Unfortunately, Yin et al. do not provide code of their implementation, so we could not have experimented with their prior based on DeepInversion.
>
> **Should Table 1 also mention Jeon et al.?**
>
> The key idea of Jeon et al. is to search for the reconstruction in the latent space of a generative model, instead of image space, so it is difficult to compare it with other attacks in Table 1. Moreover, they use StyleGAN2 for which it is not actually possible to compute the density $p(x)$.
>
> **Can you experiment with the prior that images should be in $[0, 1]$?**
>
> Yes, this was in fact proposed by the concurrent work of Geng et al. [1] who suggested adding a loss that penalizes $\ell_2$ distance between the input and copy of the input that was clipped between 0 and 1. We added an experiment in Appendix A.3 where we performed an evaluation on MNIST dataset to compare $\ell_2$ attack both with and without the suggested loss. We found that additional loss indeed obtains higher PSNR, meaning that having better prior is certainly beneficial.
>
>
> **Does $p(g|x)$ take into account pruning?**
>
> Yes, $p(g|x)$ takes into account both pruning and addition of noise: here we assume that random 50% of the elements of the gradient are set to 0, and after that we add Gaussian/Laplacian noise as before. Thus, here we can compute $p(g|x)$ exactly (as for the simple case where we had only Gaussian/Laplacian noise).
>
> **References**
>
> [1] Geng, Jiahui, et al. "Towards General Deep Leakage in Federated Learning." arXiv preprint arXiv:2110.09074 (2021).

---

> > ### Comment · Reviewer_6AUa · 2021-11-19
> > **Follow-up Questions**
> >
> > Thank you for the detailed response. I appreciate the additional experiments and clarifications. I have some follow-up questions:
> >
> > > Note that Langevin noise in Yin et al. is added to encourage exploration for the optimization process used, and our Bayesian framework is orthogonal to the actual optimization process used to maximize the final approximation of the objective. Investigating the optimization process of the final objective would certainly be an interesting item for future work.
> >
> > But doesn't the Langevin sampling from Yin et al. induce a prior in the Bayesian sense? To my understanding, the MC estimation discussed in Alg. 1 for $k=1$ updates the reconstruction x via evaluations of the gradient at $f(x+n)$ where $n$ is noise so that $x+n \in B(x,\delta)$ and $f$ is the log of the posterior. In constrast, with Langevin sampling, the noise $n$ is added to x after every gradient evaluation, so that after the first step, the update is also based on gradients at $f(x+n)$. I would love to see more clarifications on the difference between these sampling processes.
> >
> > > Note that one problem with fitting total variation this way is that we cannot compute the normalization constant for total variation, and thus we need an approach that works with unnormalized probabilities, so we used grid search. The other point of using stronger priors certainly makes sense, and we believe this is an interesting future work item. Unfortunately, Yin et al. do not provide code of their implementation, so we could not have experimented with their prior based on DeepInversion.
> >
> > I agree that the lack of code for Yin et al. is a problem. Just for reference, an implementation of the deepinversion prior in a few lines can be found at https://github.com/NVlabs/DeepInversion/blob/4b1925e469ac3ec5c7e88c004b2f227210414838/deepinversion.py#L29 [But this is just a reference for the future and I don't expect the authors to incorporate it into this work.]
> >
> > > Moreover, they [Jeon et al.] use StyleGAN2 for which it is not actually possible to compute the density .
> >
> > The same holds for TV and deep inversion though? I believe most of the priors in Table 1 don't yield computable densities.
> >
> > > Yes, this was in fact proposed by the concurrent work of Geng et al. [1] who suggested adding a loss that penalizes  distance between the input and copy of the input that was clipped between 0 and 1.
> >
> > I believe this prior (as projection during optimization) was already used in Hitaj et al, Geiping et al. and likely also Yin et al. (although this is hard to verify in the last case, due to missing code). The added experiments and interesting and helpful, thank you for adding them.

---

> > > ### Author Response · Authors · 2021-11-20
> > > **Response**
> > >
> > > Thank you for your reply and for providing reference on DeepInversion (we will experiment with it for the next revision). We provide responses to your follow-up questions below.
> > >
> > > **Could you clarify the difference between the Langevin sampling from Yin et al., and your MC sampling in Algorithm 1?**
> > >
> > > Yes, we have now added Appendix A.7 which discusses differences between the two sampling procedures. The main differences between the updates are: (i) we sample a point $x_1$ from the ball $B(x, \delta)$ uniformly at random, while Langevin update in Yin et al. samples noise $\eta$ from unit Gaussian, (ii) we compute gradient at the sampled point $x_1 \sim B(x, \delta)$, while Langevin update evaluates gradient directly at $x$, and adds noise $\eta$ to $x$ afterwards. While Yin et al. propose Langevin updates as a method to encourage exploration and diversity, our analysis shows how we can view both Monte Carlo estimation and Langevin updates as different ways to add noise to the optimization process, though currently we do not see a way to formally view Langevin updates as a Bayesian prior. We provide a more detailed comparison of the updates in Appendix A.7.
> > >
> > >
> > > **Do most of the priors in Table 1 yield non-computable densities (e.g. TV or DeepInversion)?**
> > >
> > > You are right that it is not tractable to compute *normalized* densities of priors in Table 1, but note that Equation 2 allows us to work with *unnormalized* densities for which we do not know the normalization constant. To see this, if we substitute $p(x) = \frac{\tilde{p}(x)}{Z}$ in Equation 2, where $Z$ is normalization constant for the prior and $\tilde{p}(x)$ is unnormalized probability and then multiply the entire integral by $Z$, we can notice that argmax does not change.
> > > This means that we can work with unnormalized probability densities such as TV and DeepInversion. We have now clarified this in the paper.
> > >
> > > However for GAN’s (such as StyleGAN2 used by Jeon et al.) we cannot compute neither normalized nor unnormalized density (there is some research trying to tackle this problem [1, 2]). This is in fact the reason why evaluation of the quality of GAN generated images can not be done using log-likelihood, as opposed to likelihood-based generative models such as VAE or flows.
> > >
> > > **References**
> > >
> > > [1] Eghbal-zadeh, Hamid, and Gerhard Widmer. "Likelihood estimation for generative adversarial networks." arXiv preprint arXiv:1707.07530 (2017).
> > >
> > > [2] Grover, Aditya, Manik Dhar, and Stefano Ermon. "Flow-gan: Combining maximum likelihood and adversarial learning in generative models." Thirty-second AAAI conference on artificial intelligence. 2018.

---

> > > > ### Comment · Reviewer_6AUa · 2021-11-22
> > > > **Thanks for the additional Feedback**
> > > >
> > > > Thank for this feedback. This is very helpful. I have updated my score and have no further questions.

---

### Official Review · Reviewer_DWMa · 2021-11-05

**Correctness:** 3
**Technical Novelty And Significance:** 2
**Empirical Novelty And Significance:** 2
**Recommendation:** 6
**Confidence:** 5

**Main Review:**

1) The theoretical formulation proposed in the paper is intuitive and appears to be mostly correct.

2) The key limitation of the work is the lack of much practical impact, due to the following reasons.

(a) The paper admits that for most sophisticated defense mechanisms, it is not possible to define a Bayes optimal adversary. Instead the paper falls back on designing custom attacks for each of these defense mechanisms.

(b) It is also not surprising that most existing defense mechanisms are ineffective against early stages of training, which are most sensitive to the input data. As training progresses and the models converge, the inputs are expected to have little impact on the gradients. So, this cannot be considered as a significant finding.

(c) From Table 2, it appears that the Bayes optimal attack does not produce significantly better reconstructions compared to existing attacks (l_2 and cos), except in the case where pruning is involved (last two rows). Furthermore, the paper does not claim reduction in the computational complexity of the attack using the Bayes optimal attack. Thus, there appears to be no major benefit from using the Bayesian framework.

(d) The ablation study (Figure 4) shows that even for the simplest distributions, mis-specification of the prior and conditional distributions can lead to sub-optimal reconstruction. For real-world inputs, it may be simply impossible to precisely define the prior and conditional distributions to carry out a Bayes optimal attack.

**Summary Of The Paper:**

The paper proposes a theoretical Bayesian framework for the problem of gradient leakage in federated learning. It shows that recent gradient leakage attacks are approximations of the Bayesian framework with different prior distributions for the input and conditional distributions of the gradients given the input. The paper also claims that recent defense mechanisms are not good enough, especially during the early stages of training.

**Summary Of The Review:**

The proposed Bayesian formulation for gradient leakage is reasonable, but has little practical significance because it neither provides an easier way to break sophisticated defense mechanisms nor does it lead to better reconstructions. The design of attacks against recent defense mechanisms (Soteria, ATS, and Precode) may be a useful contribution, but does not merit a publication in itself.

Since the authors have managed to show that the proposed attacks can be derived as approximations of the Bayes optimal adversary and are not purely custom designed, I have updated the rating.

---

> ### Author Response · Authors · 2021-11-14
> **Response to Reviewer DWMa**
>
> Thank you for your feedback and suggestions. We address your questions below, and we also provide a response to the common question raised by several reviewers in a separate comment.
>
> **Is your finding that most existing defense mechanisms are ineffective in early stages of training surprising?**
>
> Yes, we believe this finding is surprising and non-trivial, and was missed by prior work which evaluated defenses only at the end of training. It is important for the community to be aware of this vulnerability so that we can create stronger defenses in the future.
>
> **Is there a benefit of the Bayesian framework compared to the existing attacks, based on the results from Table 2?**
>
> Yes, note that for the Gaussian defense, $\ell_2$ and Bayes are equivalent (see last paragraph in Section 4) so they naturally achieve similar results, and in all other cases Bayes optimal attack achieves (often significantly) better reconstruction PSNR. PSNR is logarithmic, so small differences in PSNR can in fact lead to large differences in perceived quality of reconstructions. As soon as a defense produces a distribution of gradients that is different from the Gaussian, there is significant benefit in using the Bayes optimal attack, as our experiments show. Note that our goal was not to minimize the computational complexity, as any defense has to protect against an attack of any complexity (e.g. attacker could store weights and gradients, and then reconstruct the input later in the offline setting using more resources).
>
> **Is Bayes optimal attack useful given that we cannot compute prior and conditional distribution in practice?**
>
> In practice, an attack can succeed even in the case when the prior is only an approximation, as shown in Table 2 -- where all attacks use total variation prior, which is different from the true underlying image prior which is indeed not available. We agree that using the theoretically optimal Bayes attack is often not possible in practice, but we can still run the attack with these previously mentioned approximations. Moreover, we performed an additional experiment (see Appendix A.3) which shows that even a simple image prior that pushes pixel values between 0 and 1 can improve the results. Given that we showcased the problems of existing defenses, we believe it is important to consider stronger attacks for evaluation.
>
> **Does design of attacks against recent defenses merit a publication?**
>
> Yes, we believe so. Our investigation of defenses against gradient leakage is inspired by a long line of work at the intersection of machine learning and security which designs attacks against defenses for DeepFake-image detectors [1], privacy-preserving encoding [2], defenses against adversarial examples [3] (best paper at ICML 2018). We believe that our proposed attacks can lead to better defenses against gradient leakage in the future, and that our contribution indeed merits a publication.
>
> **References**
>
> [1] Carlini, Nicholas, and Hany Farid. "Evading deepfake-image detectors with white-and black-box attacks." Proceedings of the IEEE/CVF Conference on Computer Vision and Pattern Recognition Workshops. 2020.
>
> [2] Carlini, Nicholas, et al. "NeuraCrypt is not private." arXiv preprint arXiv:2108.07256 (2021).
>
> [3] Athalye, Anish, Nicholas Carlini, and David Wagner. "Obfuscated gradients give a false sense of security: Circumventing defenses to adversarial examples." International conference on machine learning. PMLR, 2018.

---

> > ### Comment · Reviewer_DWMa · 2021-11-19
> > **Response to Author Rebuttal**
> >
> > 1) The work of Geiping et al. [2020] has already shown that it is easier to attack untrained networks. Therefore, it is quite natural that even after introducing a defense mechanism, untrained networks would remain more vulnerable. In particular, for differential privacy-based defense mechanisms, if the noise variance is fixed, gradients from early stages of training are more vulnerable due to their the higher magnitude.
> >
> >
> > 2) In the response, it has been claimed that:
> >
> > "Sections 5 and 6 present a more practical viewpoint where we evaluate existing defenses in practice using attacks that are different approximations of the optimal adversary (these approximations are described in Section 4)}".
> >
> > While Section 4 does show that existing attacks can be modeled as approximations of the Bayes optimal adversary, the key question is whether the new attacks in Section 5 against current defenses are also approximations of the Bayes optimal adversary. In fact, the paper states that:
> >
> > "While in Section 4 we have shown that Bayes optimal adversary is the optimal attack for any defense (each inducing different p(g|x)), computing it is not tractable for the three defenses we consider. Here, it is more efficient to create a __custom attack__ tailored to each defense, which is enough to break their privacy promise."
> >
> > If it can be shown that the attacks in Section 5 are approximations of the Bayes optimal adversary, then the proposed framework is indeed useful. On the other hand, if the attacks in Section 5 are "custom attacks" that are not directly derived from the Bayes optimal adversary, then the impact of the proposed framework becomes questionable. Such custom attacks must be evaluated on their own merits/demerits and not clubbed to the rest of the proposed framework.

---

> > > ### Author Response · Authors · 2021-11-20
> > > **Response**
> > >
> > > Thank you for your reply. We provide responses to your concerns below.
> > >
> > > **Does DP (differential privacy) suffer from the same problem as other defenses where gradients in the early stage are more vulnerable due to the higher magnitude? Is it natural that untrained networks remain vulnerable even after introducing the defense mechanism?**
> > >
> > > No, DP does not actually suffer from this problem because it performs clipping of the gradients (see Algorithm 1 in Abadi et al. [1]) which allows it to provide formal privacy guarantees that hold regardless of the magnitude of the gradients (guarantees depend only on the clipping constant $C$, which can be fixed or dynamically chosen based on the desired privacy guarantees), meaning that it protects privacy even at the beginning of the training.
> > >
> > > While you are right that Geiping et al. has already shown that *undefended* networks are easier to attack at the early stages, our contribution is showing that the problem also occurs for *defended* networks, as the evaluation of the early stages is missing from the papers that introduced the respective defenses. This is the key limitation of proposed defenses, and it is not natural to expect that this limitation holds for any defense mechanism (given that DP does not have this problem).
> > > Thus, the key takeaway from our evaluation is that heuristic defenses do not offer any privacy protection (unlike DP which offers protection throughout training), and users should avoid building systems based on these particular defenses, which was not clear before our paper.
> > >
> > > **Can you show that attacks introduced in Section 5 are approximations of Bayes optimal adversary?**
> > >
> > > Yes, we have now added Appendix A.6 which explains how our attacks can be viewed as different approximations of the Bayes optimal adversary. In particular, Table 6 shows how each of these attacks corresponds to different distributions $p(x)$ and $p(g|x)$. Note that when we say that our attacks are custom, we mean they are adapted to each particular defense, as is suggested in the literature (e.g. see Tramer et al. [2]).
> > >
> > > **References**
> > >
> > > [1] Abadi, M., Chu, A., Goodfellow, I., McMahan, H. B., Mironov, I., Talwar, K., & Zhang, L. (2016, October). Deep learning with differential privacy. In Proceedings of the 2016 ACM SIGSAC conference on computer and communications security (pp. 308-318).
> > >
> > > [2] Tramer, F., Carlini, N., Brendel, W., & Madry, A. (2020). On adaptive attacks to adversarial example defenses. arXiv preprint arXiv:2002.08347.

---

> > > > ### Comment · Reviewer_DWMa · 2021-11-22
> > > > **Response**
> > > >
> > > > Based on the above discussion and the corresponding update by the authors, I have updated my rating.

---

### Author Response · Authors · 2021-11-14
**General response**

We thank all the reviewers for their thoughtful feedback and suggestions. We are glad that reviewers found our paper interesting and well-written. Here we provide a response to the common question that has arised in several reviews, and then separately answer individual questions.

**Could you clarify the connection between Bayesian framework (Section 4) and attacks on existing defenses (Sections 5, 6)?**

The central challenge that our paper tackles is how to appropriately evaluate defenses against gradient leakage. To this end, Section 4 approaches the task from a theoretical standpoint, and establishes a theoretically optimal attack for each defense. Sections 5 and 6 present a more practical viewpoint where we evaluate existing defenses in practice using attacks that are different approximations of the optimal adversary (these approximations are described in Section 4). In that sense, the two parts of our paper are both valuable as they complement each other by providing both theoretical and practical insights on evaluating defenses. We have now clarified this connection in Section 1.

---

### Decision · Program_Chairs · 2022-01-20

**Decision:**

Accept (Poster)

**Comment:**

The paper formalizes the problem of gradient leakage through a Bayesian framework. They show that existing attacks can be viewed as approximations of a Bayesian optimal adversary. The empirical results show that heuristic defences are not good against stronger attacks and that the early part of the training is particularly vulnerable. There was a lively discussion in the reviews and rebuttal and the outstanding questions of the reviewers have been answered.